# PATH SPACE FOR RECURRENT NEURAL NETWORKS WITH ReLU ACTIVATIONS

## ABSTRACT

It is well known that neural networks with rectified linear units (ReLU) activation functions are positively scale-invariant (i.e., the neural network is invariant to positive rescaling of weights). Optimization algorithms like stochastic gradient descent that optimize the neural networks in the vector space of weights, which are not positively scale-invariant. To solve this mismatch, a new parameter space, called path space, has been proposed for feedforward and convolutional neural networks. The path space is positively scale-invariant and optimization algorithms operating in path space have been shown to be superior than that in the original weight space. However, the theory of path space and the corresponding optimization algorithm cannot be naturally extended to more complex neural networks, like Recurrent Neural Networks (RNN) due to the recurrent structure and the parameter sharing scheme over time. In this work, we aim to construct path space for RNN with ReLU activations so that we can employ optimization algorithms in path space. To achieve the goal, we propose leveraging the reduction graph of RNN which removes the influence of time-steps, and prove that all the values of paths in the reduction graph can serve as a sufficient representation of the RNN with ReLU activations. We then prove that the path space for RNN is composed by the basis paths in reduction graph, and design a *Skeleton Method* to identify the basis paths efficiently. With the identified basis paths, we develop the optimization algorithm in path space for RNN models. Our experiments on several benchmark datasets show that we can obtain more effective RNN models in this way than using optimization methods in the weight space.

## 1 INTRODUCTION

Over the past ten years, ReLU activations have become increasingly popular in various types of neural networks such as Multilayer Perceptron(MLP) (Nair & Hinton, 2010; Glorot et al., 2011; Neyshabur et al., 2015a) , Convolutional Neural Networks (CNN) ((Krizhevsky et al., 2012; He et al., 2016)) and Recurrent Neural Networks (RNN) (Le et al., 2015; Neyshabur et al., 2016). Theoretical studies on the MLP and CNN with ReLU activations, particularly on its positively scale-invariant (PSI) property have also been conducted (Neyshabur et al., 2016; 2015a; Dinh et al., 2017; Meng et al., 2018). Specifically, PSI property means if the incoming weights of a hidden node (or a feature map for CNN) are multiplied by a positive scalar $c$, and the outgoing weights of this hidden node (or the feature map) are divided by $c$, the output for arbitrary input will keep unchanged. However, conventional algorithms like stochastic gradient descent optimize the neural networks in the vector space of weights, which are not positively scale-invariant. This mismatch may lead to problems during the optimization process (Neyshabur et al., 2015a; Meng et al., 2018), e.g., in an unbalanced network (i.e., the norm of incoming weights to different units have relatively large differences), the training process of vanilla stochastic gradient descent will be significantly slowed down.

To solve the above mismatch, some recent studies have been conducted on *paths* of ReLU neural networks (Neyshabur et al., 2015a; Meng et al., 2018). The value of a path is defined as the multiplication of weights along the path which starts from an input node, successively crosses a hidden node at every layer, and finally ends at an output node. Obviously, the value of a path is positively scale-invariant. Thus, a new parameter space, called path space, for MLP and CNN have been proposed. Path space is a vector space constituted by the values of given paths called *basis paths*, whose values are sufficient to calculate the output of the MLP and CNN with ReLU activations.

In a word, path space is a PSI parameter space and can sufficiently represent MLP and CNN with ReLU activation. In the work (Meng et al., 2018), an algorithm is designed to identify the basis paths for MLP and CNN. Efficient optimization algorithms in path space have also been proposed, which iteratively update the values of basis paths according to the gradient of loss with respect to basis paths. Optimizing MLP or CNN with ReLU activations in this way can achieve better performances compared with traditional optimization methods in the weight space.

However, the studies of path space especially the identification of basis paths depend on the network structure, so it cannot be naturally extended to more complex network structures, like recurrent neural network (RNN). RNN is a powerful neural network model to process sequential data and show excellent performance in various domains such as language modeling(Mikolov et al., 2010; Jozefowicz et al., 2016), machine translation(Bahdanau et al., 2014) and speech recognition (Graves et al., 2013; Miao et al., 2015). RNN with ReLU activations (abbrev. ReLU RNN) is also positively scale-invariant (Neyshabur et al., 2016). Considering the superiority of optimizing MLP or CNN with ReLU activations in their path space, the path space for RNN is worth studying.

In this work, we aim to construct path space for ReLU RNN so that we can employ the optimization algorithm in path space. Due to the recurrent structure, the paths in RNN depend on the time-steps. That means, if the sequence length of the input is not pre-given, the number of paths in RNN cannot be determined. It makes the identification of basis paths from the undetermined number of paths difficult. To handle this, we propose leveraging the reduction graph of RNN to remove the influence of time-steps and theoretically develop path space for RNN in the following three steps. First, we define paths on reduction graph, whose number is fixed and not influenced by the sequence length. Second, we prove that the paths on reduction graph can serve as a sufficient representation of the RNN with ReLU activations. Third, we define path space for RNN which is composed by the values of basis paths in the reduction graph.

Next, we design *Skeleton Method* to identify basis paths in reduction graph of RNN efficiently. Compared with MLP and CNN, the skeleton method for RNN needs extra steps to handle the recurrent weights. Specifically, the skeleton method for RNN contains two steps: it first identifies the basis paths in the reduction graph of RNN without recurrent weights; then it constructs the basis paths that contain recurrent weight. We prove that paths selected by *Skeleton Method* are basis paths, and the number of basis paths which is also the dimension of the path space is #(weights)−#(hidden nodes) in the reduction graph of RNN.

Finally, with the identified basis paths, we employ optimization algorithm in path space for RNN models, which operates update on basis paths according to the gradient of loss w.r.t basis paths. Our experiments on several benchmark datasets show that we can obtain significantly more effective RNN models in this way than using optimization methods in the weight space. It indicates that optimizing ReLU RNN in path space can solve the optimization problem brought by positively scale-invariant property and can indeed help optimization. To the best of our knowledge, our work is the first to construct the new parameter space, path space, for the RNN model and the first to optimize RNN in its path space.

The paper is organized as follows: in Section 2, we introduce the background about the path representation of ReLU RNN and related works; in Section 3, we formally define the basis path for ReLU RNN and propose using reduction graph as a tool to handle the difficulty when identifying the basis paths in ReLU RNN; in Section 4, we present our the basis path identification algorithm and the optimization algorithm in the path space; in Section 5, we show the experiment results to verify the effectiveness of our proposed algorithm; finally, we conclude this paper in Section 6.

## 2 BACKGROUND

In this section, we briefly review the path representation of recurrent neural networks with ReLU activations and the related works in this direction.

### 2.1 PATH REPRESENTATION OF ReLU RNN

As shown in Figure 1, the network structure of RNN can be regarded as a **directed graph** $G(\mathcal{N}, \mathcal{E})$, where $\mathcal{N} = \cup_t \mathcal{N}^t$ denotes the set of nodes and $\mathcal{E}$ denotes the set of edges, where

$\mathcal{N}^t = \{N_1^{I,(t)}, N_2^{I,(t)} \cdots, N_{d+1}^{H,(t)}, N_{d+2}^{H,(t)}, \cdots, N_{d+h+1}^{O,(t)}, N_{d+h+2}^{O,(t)} \cdots N_{d+h+K}^{O,(t)}\}$ denotes the set of nodes at the $t$-th time-step, the superscripts $I, H, O$ means the input, hidden and output node, respectively. Here, $d, h, K$ denote the dimension of input, hidden nodes and output, respectively, and the superscript $t$ denotes the time-step. In the graph $G(\mathcal{N}, \mathcal{E})$, there must be an edge between an input node and a hidden node that are at the same time step, a hidden node and an output node that are at the same time-step, and two hidden nodes whose time-steps are adjacent. A **Path** on the $G(\mathcal{N}, \mathcal{E})$ is defined in Definition 1.

**Definition 1 (Paths in RNN).** *A path in the recurrent neural network is defined as a list of nodes and the corresponding edges between adjacent nodes, which starts from an input node $N_{i_0}^{I,(t)}$ to one output node $N_{i_{s+1}}^{O,(t+s)}$ with $s \geq 0$ by successively crossing hidden nodes. Specifically, a path can be expressed as $N_{i_0}^{I,(t)} \rightarrow N_{i_1}^{H,(t)} \rightarrow \cdots \rightarrow N_{i_{s+1}}^{H,(t+s)} \rightarrow N_{i_{s+2}}^{O,(t+s)}$, and there must be an edge between the adjacent nodes, where $i_{\cdot}$ is the index of node along the path.*

For example, the red line in Figure 1 shows an example of the paths. For simplicity, we omit the explicit expression of paths and use $p$ to denote the index of paths in the following context. **The value of a path** is defined as the multiplication of the values of weights on edges contained in the path. Specifically, for an edge $e_k \in \mathcal{E}$, we use $w_k$ to denote the weight on edge $e_k$, then the value of path $p$ is $v_p = \prod_{e_k \in p} w_k$.

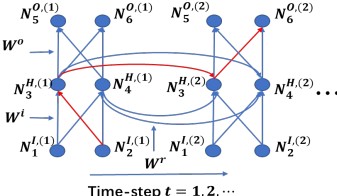

Figure 1: DAG of RNN

Using $W^i, W^o$ and $W^r$ to denote three weight matrices, use $x^{(t)}$ to denote the input and use $n^{H,(t)}, n^{O,(t)}$ to denote the corresponding hidden and output values at nodes $N^{H,(t)}$ and $N^{O,(t)}$, recurrent neural network computes the output vector $n^{O,(t)}$ at $t$-th time-step by a recurrent formulation as:

$$n^{H,(t)} = \sigma(W^i x^{(t)} + W^r n^{H,(t-1)}), \tag{1}$$

$$n^{O,(t)} = W^o n^{H,(t)}. \tag{2}$$

If the activation $\sigma(\cdot)$ is ReLU, i.e., $\sigma(z) = \max(z, 0)$, we can get $n^{O,(t)} = W^o \cdot (W^i x^{(t)} + W^r n^{H,(t-1)}) \cdot diag(\mathbb{I}(n_1^{H,(t)} > 0), \cdots, \mathbb{I}(n_h^{H,(t)} > 0))$ by combining Eq.(1) and Eq.(2). [1] According to the definition of value of a path, the output of ReLU RNN can be calculated using the values of paths and its activation status as follows,

$$n_k^{O,(T)}(x) = \sum_{i=1}^{d} \sum_{t=1}^{T} \sum_{p \in \mathcal{P}_{T,k,i,t}} v_p \cdot a_p(w; x) \cdot x_i^{(t)}, \tag{3}$$

where the set $\mathcal{P}_{T,k,i,t}$ consists of the paths that connects the $k$-th output at time-step $T$ and the $i$-th input at time-step $t$ with length $T + 1 - t$, $v_p$ is the value of path $p$, and $a_p(w; x) = \prod_{N_i^{H,(t)} \in p} \mathbb{I}(n_i^{H,(t)}(w; x) > 0)$ is the activation status of path $p$.

## 2.2 RELATED WORKS

In recent years, researchers have started to conduct theoretical studies on optimization and generalization of ReLU neural network by leveraging paths instead of weights. In the aspect of generalization, the works (Neyshabur et al., 2015b; 2017; Zheng et al., 2018) show that the generalization error of feedforward ReLU network is related to the path-norm. The work (Weinan et al., 2019) also leverages the path representation to analyze the generalization error of ReLU neural networks. Except for the generalization analysis, optimization algorithms based on paths are also designed. Neyshabur, et al. propose Path-SGD algorithm, which optimizes the loss function regularized by path-norm (Neyshabur et al., 2015a; 2016). Path-SGD algorithm does not directly optimize ReLU RNN using the loss function composed by the path-represented output but optimizes the regularized loss defined in the original weight space. The work (Meng et al., 2018) proposes $\mathcal{G}$-SGD algorithm to optimize the ReLU MLP and CNN directly by updating the values of paths according to the gradient of loss w.r.t basis paths. However, $\mathcal{G}$-SGD is only designed for MLP and CNN. To the best of our knowledge, the path space constituted by basis paths for ReLU RNN has not been studied yet. In this paper, we will investigate the path space for RNN so that we can apply $\mathcal{G}$-SGD to RNN.

---

[1] Here, $diag(a_1, \cdots, a_h)$ denotes a diagonal matrix whose diagonal elements are $a_1, \cdots, a_h$.

## 3 PATH SPACE FOR RELU RNN

In this section, we investigate the path space, a vector space constituted by values of basis paths for ReLU RNN. We first formally define the basis paths for ReLU RNN. Then we propose using reduction graph as a tool to define the path space for ReLU RNN.

### 3.1 DEFINITION OF BASIS PATHS FOR RNN

As stated in the introduction, it has been shown in (Neyshabur et al., 2016) (cf. Theorem 1 and Figure 1) that the output of RNN is positively scale-invariant (PSI) i.e., if all the incoming weights(including the shared weights) of a hidden node are multiplied by a positive scalar $c$ and all the outgoing weight (including the shared weights) of this hidden node are multiplied by $1/c$, ReLU RNN will generate the same output for arbitrary input. Because the values of paths are also positively scale-invariant, which matches the PSI property of ReLU RNN, we are motivated to directly represent and optimize the RNN model according to the path representation (Eq.(3)).

However, we cannot directly optimize the values of all the paths by regarding each of them to be an independent parameter, because they are correlated with each other. We use Figure 1 to show an example. For simplicity, we denote the weight of the edge that connects node $i$ and node $j$ as $w_{ij}$.[2] We denote the red path in Figure 1 as $p_1 : N_2^{I,(1)} \rightarrow N_3^{H,(1)} \rightarrow N_3^{H,(2)} \rightarrow N_6^{O,(2)}$ whose value is $v_{p_1} = w_{23} \cdot w_{33} \cdot w_{36}$. We can show that the value of path $p_1$ can be calculated by the values of following four paths through multiplication and division.

$$p_2 : N_2^{I,(1)} \rightarrow N_3^{H,(1)} \rightarrow N_5^{O,(1)} \qquad\qquad v_{p_2} = w_{23} \cdot w_{35}$$

$$p_3 : N_1^{I,(1)} \rightarrow N_3^{H,(1)} \rightarrow N_3^{H,(2)} \rightarrow N_5^{O,(1)} \qquad v_{p_3} = w_{13} \cdot w_{33} \cdot w_{3,5}$$

$$p_4 : N_1^{I,(1)} \rightarrow N_3^{H,(1)} \rightarrow N_6^{O,(1)} \qquad\qquad v_{p_4} = w_{13} \cdot w_{36}$$

$$p_5 : N_1^{I,(1)} \rightarrow N_3^{H,(1)} \rightarrow N_5^{O,(1)} \qquad\qquad v_{p_5} = w_{13} \cdot w_{35}.$$

It is easy to verify that

$$v_{p_1} = \frac{v_{p_2} \cdot v_{p_3} \cdot v_{p4}}{\left(v_{p_5}\right)^2}. \tag{4}$$

Because $v_{p_1}$ can be calculated using $v_{p_2}, \cdots, v_{p_5}$, we only need to know $v_{p_2}, \cdots, v_{p_5}$ and the relation in Eq.(4) when to represent the output of RNN.

For any given recurrent neural network, we need to first investigate and decouple the correlation among their values so that we can identify a subset of paths whose values can be directly optimized. Motivated by the example, we formally define basis paths as follows.

**Definition 2 (Basis Paths).** *A set of paths $\mathcal{P}_b$ are called basis paths if they satisfy the following properties: (1) for any path $p \in \mathcal{P}/\mathcal{P}_b$, there exists non-zero coefficient vector $\alpha = (\alpha_1, \cdots, \alpha_z)$ to make $v_p(w) = \prod_{j \in \mathcal{P}_b} v_j(w)^{\alpha_j}$ for any $w \in \mathcal{W}$, where $z$ denotes the cardinal number of $\mathcal{P}_b$, i.e., $z = |\mathcal{P}_b|$; (2) among all the sets that satisfy property (1), $\mathcal{P}_b$ has the smallest cardinal number.*

The first item in the definition ensures the sufficiency to use basis paths to calculate the values of other paths, and the second item ensures that the values of basis paths are independent, i.e., the values of basis paths cannot be calculated by each other. In the work (Meng et al., 2018), basis paths have been defined for MLP, however, it is no longer suitable for RNN. Here, the definition 2 for basis paths is more general and can be generalized to many kinds of network structures.

### 3.2 DEFINING PATH SPACE FOR RELU RNN THROUGH REDUCTION GRAPH

Although basis paths have been defined in the previous section, given an RNN, it is difficult to identify the basis paths because: (1) the number of paths exponentially depends on the time-steps, which is very large in general; (2) if the sequence length of input is not pre-given, the exact number of paths in RNN cannot be determined. To handle this, we study the paths in reduction graph of ReLU RNN.

Here, we give the definition of the reduction graph for RNN and show an example in the Figure 2.

---

[2]The weights are shared for all time-steps, so we don't use the subscript of time-step $t$ in the weight matrices.

**Definition 3** (**Reduction Graph**). *Given a directed graph of RNN, we use $G^r(\mathcal{N}^r, \mathcal{E}^r)$ to denote its corresponding reduction graph, where $\mathcal{N}^r = \{N_1^I, N_2^I \cdots, N_{d+1}^H, N_{d+2}^H, \cdots, N_{d+h+1}^O, N_{d+h+2}^O \cdots N_{d+h+K}^O\}$ denotes the set of nodes and $\mathcal{E}^r$ denotes the set of edges. There must be an edge between an input node and a hidden node, a hidden node and an output node and two hidden nodes, respectively. We call the edges that connect two hidden nodes as recurrent edges.*

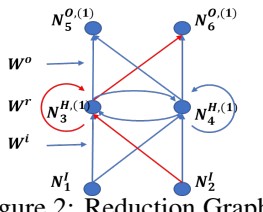

Figure 2: Reduction Graph of RNN

**Definition 4** (**Path in Reduction Graph**). *A path in the reduction graph for RNN is a list of nodes that satisfies the following conditions: (1) it starts from an input node and end at an output node of the reduction graph; (2) it can at most cross one recurrent edge.*

According to the definition, the paths in reduction graph are classified into two categories: the paths does not contain recurrent edge denoted as $N_{i_0}^I \to N_{i_1}^H \to N_{i_2}^O$; contain one recurrent edge denoted as $N_{i_0}^I \to N_{i_1}^H \to N_{i_2}^H \to N_{i_3}^O$. Hence, the paths in reduction graph is not related to the time-steps. That means, no matter how long the sequence length of the input, the number of paths in reduction graph is fixed. The next theorem shows that the values of paths in the reduction graph are sufficient to represent RNN.

**Theorem 1.** *All the values of paths in directed graph of RNN can be calculated using the values of paths in the reduction graph of RNN through multiplication and division operators.*

We put the proof of Theorem 1 in Appendix and only deliver the techniques used in the proof here. We index the trained weights as $w_1, \cdots, w_i, \cdots, w_m$, and represent each path using an $m$-dimensional vector $c_p = (c_{p1}, \cdots, c_{pm})$ where $c_p$ satisfy $v_p = \prod_{i=1}^m w_i^{c_{pi}}$. We prove that the vectors of all paths can be calculated through linear combinations of vectors of paths in reduction graph. Thus, we can get the result in Theorem 1.

We use $\mathcal{P}$ to denote the set constituted by all paths of RNN, $P_r$ to denote the set constituted by paths in reduction graph and $\mathcal{P}_b$ to denote the set constituted by basis paths of RNN. According to Definition 2 and Theorem 1, we have $\mathcal{P}_b \subset \mathcal{P}_r \subset \mathcal{P}$. Thus, the basis paths in the reduction graph of RNN is the basis paths of RNN model. Paths in reduction graph of RNN is totally determined by the model and not influenced by the time-steps, and as well as the basis paths.

Next, we show that basis paths are sufficient to represent the output of ReLU RNN. According to Eq.(3) and item (1) in the definition of basis paths, we only need to prove that the activation status of paths can also be determined by the values of basis paths.

**Theorem 2.** *The activation status of the paths in ReLU RNN can be calculated using the values of basis paths and the signs of the selected elements in $W^o$ if for each column in matrix $W^o$ with dimension $K \times h$, we randomly select one element whose sign is pre-given and fixed.*

Using the basis paths, we can construct a new vector space, called path space, for ReLU RNN as :

$$V := \{v = (v_{p_1}, \cdots, v_{p_z}) : v \in (\mathbb{R}/\{0\})^z\}, \tag{5}$$

where $p_i, i = 1, \cdots, z$ are the basis paths for RNN.

Till now, we have defined path space for ReLU RNN and proved that path space is sufficient to represent ReLU RNN in Theorem 2. In next section, we will introduce how to optimize ReLU RNN in its path space including that how to identify basis paths for given reduction graph.

## 4 OPTIMIZATION ALGORITHM IN PATH SPACE FOR RNN

In this section, we design *Skeleton Method* to identify the basis paths in reduction graph of RNN efficiently. Then, we introduce the update rule of optimization algorithm in path space for RNN.

We decompose the reduction graph of recurrent neural network into two parts: the feedforward part and the recurrent edges part. The feedforward part contains all nodes in reduction graph and the non-recurrent edges, and the recurrent edges part contains only the recurrent edges.

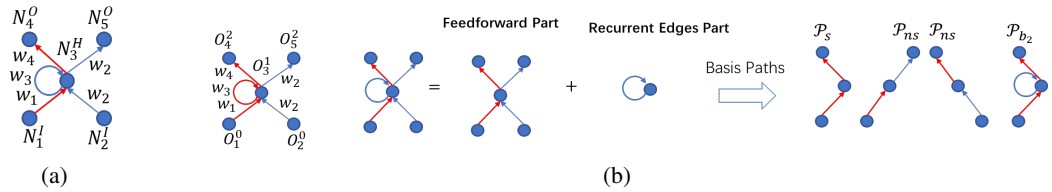

Figure 3: (a) An example of reduction graph.    (b) An example of *Skeleton Method*

In *Skeleton Method*, we first define the *skeleton edges* in the feedforward part and then use the skeleton edges to construct the basis paths. We prove that the paths that contains at most one *non-skeleton edges* are basis path.

The *Skeleton Method* for RNN is described in the following box.

---

*Skeleton Method*

(1) For each hidden node in the feedforward part, we randomly select one edge that pointing to this hidden node and randomly select one edge that starts from this hidden node. Putting the selected edges of all hidden nodes together, all the selected edges are called skeleton edges. All other edges are called *non-skeleton edges*.

(2) The paths that contain at most one non-skeleton edges in the feedforward part are selected into set $\mathcal{P}_{b_1}$.

(3) The paths containing only skeleton edges and one recurrent edge are selected into set $\mathcal{P}_{b_2}$.

(4) The final selected path set is $\mathcal{P}_{b_1} \cup \mathcal{P}_{b_2}$.

---

We illustrate the *Skeleton Method* in Figure 3(b). The red edges are the skeleton edges selected in step (1). Left three basis paths are in $\mathcal{P}_{b_1}$ and the last one is in $\mathcal{P}_{b_2}$. For the paths in $\mathcal{P}_{b_1}$, we further denote the set constituted by paths only contain skeleton edges as $\mathcal{P}_s$ and the set constituted by paths contain one non-skeleton edges as $\mathcal{P}_{ns}$.

In the following theorem, we prove that the paths selected by *Skeleton Method* are basis paths.

**Theorem 3.** *The set of paths $\mathcal{P}_{b_1} \cup \mathcal{P}_{b_2}$ selected by the Skeleton Method are basis paths. The number of basis paths (or equivalently the dimension of the path space) for recurrent neural networks is #(weights)−#(hidden nodes), where "#" means "the number of".*

After identifying the basis path using *Skeleton Method*, we aim to optimize the loss function $L(v)$ by following the negative direction of the gradients of loss w.r.t basis paths as below,

$$v_{p_i}^{t+1} = v_{p_i}^t - \eta_t \cdot \frac{\partial L(v)}{\partial v_{p_i}}\Big|_{v=v_{p_i}^t}, \tag{6}$$

where $p^i$ is the index of the basis paths. $\mathcal{G}$-SGD algorithm has been designed in work (Meng et al., 2018), which can efficiently implement the above update rule in path space for MLP. Here, we apply $\mathcal{G}$-SGD to RNNs. Due to space limitation, we only introduce the high-level idea as below and the update rule in Alg. 2. For more details about its derivation, please refer to the Appendix B.1. or Eq.(4) and Eq.(5) in (Meng et al., 2018).

**First**, we construct mask matrices to distinguish the skeleton edge and non-skeleton edge according to step (1) in skeleton method. As shown in Alg.1, it generates mask matrices $M^i, M^o$ which have the same size with $W^i$ and $W^o$, respectively, and using elements 1 and 0 in the mask matrices to denote skeleton and non-skeleton edge, respectively. **Second**, $\mathcal{G}$-SGD calculates the gradient w.r.t basis path by using the fact that the value of basis path is multiplication of weights on this path. So it first calculates the gradient w.r.t weights (Line 3-4 in Alg.2) and then using the relation between paths and weights to obtain the gradients w.r.t basis paths $G_{p_s}, G_{p_{ns}^o}$ and $G_{p_{ns}^r}$ where footnotesize$p_s \in \mathcal{P}_s$, $p_{ns}^o \in \mathcal{P}_{ns}, p_{ns}^r \in \mathcal{P}_{b_2}$ (Line 6 in Alg.2). This step is called Inverse-Chain-Rule. **Third**, after updating the basis paths according to Eq.(6), it calculates the ratio $R_i = v_{p_i}^{t+1}/v_{p_i}^t$ for $p_i \in \mathcal{P}_s$. **Fourth**, it allocates the ratio to weights to obtain new weights (Line 9-10 in Alg.2) in order to implement forward process in next iteration[3]. Detailed derivation can be referred to Appendix B.1.

---

[3]Operations such as $shape(\cdot), zero\_like, .sum(0)$ in Alg. 1 and 2 is similar to the function in the numpy packages of python language. The exact definition are listed in Appendix B.2.

---

**Algorithm 1:** Mask-Matrix-Constructor

---

**Input**: weight matrices of RNN model $W^i, W^o$
$M^i = zeros\_like(W^i)$,
$M^o = zeros\_like(W^o)$ ;
$n\_out = \text{shape}(W^i)[0]$ ; $n\_in = \text{shape}(W^i)[1]$ ;
**for** $j = 1$ **to** $n\_out$ **do**
$\quad\mid\quad M^i[j][j \% n\_in] = 1$ ;
**end**
$n\_out = \text{shape}(W^o)[0]$ ; $n\_in = \text{shape}(W^o)[1]$ ;
**for** $j = 1$ **to** $n\_in$ **do**
$\quad\mid\quad M^o[j][j \% n\_out] = 1$ ;
**end**
**Output**: Mask Matrix $M^i, M^o$

---

**Algorithm 2:** $\mathcal{G}$-SGD for RNN

---

**Input**: RNN model $F_{RNN}(W^i, W^r, W^o)$, data $X, Y$, loss function $loss\_func$, learning rate $\eta$
1  $M^i, M^o = $ *Mask-Matrix-Constructor($W^i, W^o$ )* ;
2  **repeat**
3  $\quad$ Sample a batch of data $x, y$, compute the loss function $loss = loss\_func(F_{RNN}(x), y)$ ;
4  $\quad$ Using Back-Propagation to compute the gradient w.r.t the weight:
$\quad\quad G_{W^i}, G_{W^r}, G_{W^o} = BP(loss, W^i, W^r, W^o)$ ;
5  $\quad$ **### Using Inverse-Chain-Rule to obtain the gradient w.r.t basis paths:**
6  $\quad G_{p_s} = G_{W^i} - \frac{(W^r \cdot G_{W^r}).sum(0) + (W^o \cdot G_{W^o}).sum(0)}{(W^i \cdot M^i).sum(0)}$; $\quad G_{p_{ns}^o} = \frac{G_{W^o}}{W^o}$; $\quad G_{p_{ns}^r} = \frac{G_{W^r}}{W^r}$ ;
7  $\quad$ **### Calculate the Update Ratio:** $\quad R = 1 - \eta \cdot \frac{G_{p_s}}{(W^i \cdot M^i).sum(0)}$ ;
8  $\quad$ **### Weight-Allocation and update weight matrices as:**
9  $\quad W^i = W^i \cdot diag(R) + (W^i - \eta \cdot G_{W^i}) \cdot (1 - M^i)$;
10  $\quad W^o = W^o \cdot M^o + \frac{W^o - \eta \cdot \frac{G_{p_{ns}^o}}{W^i}}{diag(R)} \cdot (1 - M^o)$; $\quad W^r = \frac{W^r - \eta \cdot \frac{G_{p_{ns}^r}}{W^i}}{diag(R)}$
11  **until** *stopping criterion is met*;
**Output**: Weights of RNN model $W^i, W^r, W^o$

---

**Computational Cost Analysis.** (1)The Mask-Matrix-Calculator(Algorithm 1) only run once at the beginning. So the extra cost can be ignored. (2)As shown in the Algorithm 2, the forward and backward phase in $\mathcal{G}$-SGD for RNN is the same as traditional SGD algorithm (Line 3- 4). The extra computational cost is in Inverse-Chain-Rule and Calculate the update ratio(Line 5-7). Please note that we only need to calculate the ratio $R$ for basis paths in $\mathcal{P}_s$ whose number equals to the number of hidden nodes, which is small compared with the number of weights. Thus the computational cost of step 5-10 is much less than the backward process. In realistic experiments, the extra time cost is **less than 10%** compared to vanilla SGD. We list the running time of each algorithm in Appendix(Tab. 4).

**Remark:** The skeleton method and the update rule of $\mathcal{G}$-SGD can be generalized to stack RNN and recurrent convolutional neural networks (RCNN), which can be referred to Appendix. Besides, according to Definition 2, the selection of basis paths is not unique. Every selection can serve as a sufficient representation of the model. *Skeleton Method* only provides one selection of them. In this work, we employ $\mathcal{G}$-SGD based on this fixed selection. For other selections, the update rule of $\mathcal{G}$-SGD can also be derived similarly.

## 5 EXPERIMENT

In this section, we verify the effectiveness of the optimization algorithms in path space for ReLU RNN. We train ReLU RNN on several tasks and show that we can obtain significantly more effective RNN models in path space than using conventional optimization methods in the weight space. Specifically, we take vanilla SGD and Path-SGD (Neyshabur et al., 2016) to be our baselines.

### 5.1 SEQUENTIAL MNIST

In this section, we optimize the ReLU RNN for the Sequential MNIST (LeCun, 1998; Neyshabur et al., 2016; Le et al., 2015; Arjovsky et al., 2015; Bai et al., 2019). In Sequential MNIST, each digit image is reshaped into a sequence, turning the digit classification task into a sequence classification task

with long-term dependencies. To make the task even harder, we also use a fixed random permutation on the pixels of the MNIST digits.

MNIST dataset is download from the official site (LeCun). The sequence length is set to 28 and 98. The hidden nodes size of ReLU RNN are 100. We search the best learning rate in a reasonable range. The size of minibatch is 64. More detailed experimental settings can refer to the Appendix A.1.

Every experiment is run 3 times with different random seeds. The averaged test accuracy is shown in the Table 1. It can be observed that the classification error rates of $\mathcal{G}$-SGD algorithm on the tasks are significantly lower than that of Path-SGD and SGD. It indicates that optimizing RNN with ReLU activations in the path space can consistently obtain more effective RNN models for all settings.

Table 1: Averaged testing classification error rate(%) of the sequential MNIST experiment.

|  |  | $\mathcal{G}$-SGD | SGD | Path-SGD |
|---|---|---|---|---|
| Non-Permutation | sMNIST-28 | **1.50 $\pm$ 0.03** | 1.59 $\pm$ 0.06 | 1.75$\pm$ 0.03 |
|  | sMNIST-98 | **2.70 $\pm$ 0.09** | 3.07 $\pm$ 0.18 | 2.99$\pm$ 0.08 |
| Permutation | sMNIST-28 | **3.80$\pm$ 0.09** | 4.19$\pm$ 0.13 | 4.12 $\pm$ 0.24 |
|  | sMNIST-98 | **5.42$\pm$ 0.05** | 5.70$\pm$ 0.16 | 5.47$\pm$ 0.03 |

## 5.2 LANGUAGE MODELING

In this section, we optimize the stacked ReLU RNN for the language modeling tasks (Merity et al., 2018) with four datasets: word-level Penn Treebank(PTB) dataset (Marcus et al., 1993), Wikitext-2 dataset (Merity et al., 2016), character-level Penn Treebank(PTBc) dataset (Merity et al., 2016) and Hutter Prize(enwik8) dataset (Hutter, 2012). The first two datasets are word level and the last two datasets are character level. For all tasks, we train a stacked ReLU RNN with 4 layers. The learning rate is turned in the range $\{10, 7.5, 5, 2.5, 1, 0.75, 0.5, 0.1\}$. Performance is evaluated using the perplexity (PPL) metric for the word level datasets and bits-per-character (BPC) metric for the character level datasets. The detailed experiment settings including the description of the datasets and hyper-parameters are put into the Appendix A.2.

The results are shown in Figure 4. The perplexity of $\mathcal{G}$-SGD algorithm on the PTB and Wikitext-2 datasets are lower than that of Path-SGD and SGD. The bits-per-character metric of $\mathcal{G}$-SGD algorithm on the PTB-c and enwik8 data sets are significantly lower than that of Path-SGD and SGD. It means that $\mathcal{G}$-SGD are more effective than Path-SGD and SGD, which shows the superiority of optimization RNN with ReLU activations in the path space.

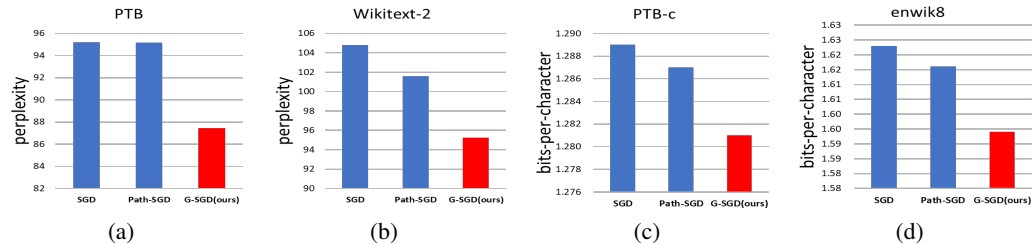

Figure 4: Test metric for language tasks. The number means PPL in PTB and Wikitext-2 datasets. The number means BPC for PTB-c and enwik8 datasets.

## 5.3 RECURRENT CONVOLUTIONAL NEURAL NETWORK

In this section, we optimize the Recurrent CNN(RCNN) for the image classification tasks(Liang & Hu, 2015). The model structure is similar to the RNN. The main difference is that the full connection in RNN is replaced by convolution layer. [4]

We evaluate the algorithms on three benchmark image classification datasets, MNIST(LeCun & Cortes, 2010), CIFAR-10 and CIFAR-100 (Krizhevsky & Hinton, 2009). For CIFAR-10 and CI-

---

[4]Path-SGD algorithm is not implemented to CNN in its original papers (Neyshabur et al., 2015a; 2016), and it is not clear how to extend it to CNN. Thus we do not use Path-SGD as a baseline for experiments on RCNN.

FAR100, we use (96,128,160) feature maps. Since MNIST is much easier, we use (32, 64, 96) feature maps. In general, We follow the experiment setup in (Liang & Hu, 2015). Most of the hyper-parameters is same as previous work such as the network structure, weight decay, scheduling of learning rate, etc . The detailed experiment setting is put in Appendix A.3.

Table 2 shows the test classification error rate. We can see that optimizing RCNN in path space outperforms the conventional optimization algorithms SGD that in weight space consistently.

Table 2: Classification error rate(%) of experiments on recurrent convolutional neural networks.

| | MNIST | | | CIFAR-10 | | | CIFAR-100 | | |
|---|---|---|---|---|---|---|---|---|---|
| | RCNN-32 | RCNN-64 | RCNN-96 | RCNN-96 | RCNN-128 | RCNN-160 | RCNN-96 | RCNN-128 | RCNN-160 |
| $\mathcal{G}$-**SGD** (our method) | **0.36** | **0.31** | **0.31** | **7.38** | **6.61** | **6.52** | **31.52** | **29.64** | **28.78** |
| SGD | 0.38 | 0.37 | 0.39 | 7.47 | 7.53 | 7.54 | 33.34 | 31.51 | 29.67 |

## 6 CONCLUSION

In this paper, we construct a new parameter space, called path space, for the RNN with ReLU activations and employ optimization algorithms in path space on benchmark experiments. To achieve this, we propose reduction graph method to deal with the difficulty brought by the parameter-sharing scheme, and propose a new *Skeleton Method* to identify the basis path for RNN in reduction graph efficiently. We conduct several experiments to verify that we can obtain significantly more effective RNN models in path space than using conventional optimization methods in the weight space. We would like to highlight the following conclusions: (1) **Basis paths are crucial in the ReLU RNN.** We have proved that the values of basis paths serve as a sufficient representation of ReLU RNN. Compared with the weights, the values of basis paths are positively rescaling-invariant, which match the PSI property of the output of ReLU RNN. Moreover, the number of basis paths is even less than the number of weights of the recurrent neural network. (2) **Reduction graph is a powerful tool.** Reduction graph and its paths are introduced to simplify the identification of basis paths in ReLU RNN. It is a novel technique specially designed for RNN models. (3) **Optimizing ReLU RNN in path space is efficient.** The algorithm $\mathcal{G}$-SGD can achieve better performances than SGD with only a little extra computational cost. In the future, we will investigate the power of path view on other neural network structures such as LSTM and Transformer.

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

APPENDIX

# A EXPERIMENT SETTINGS

All the experiments are implemented by the Pytorch framework. And all the experiments run in the Nvidia GPU.

## A.1 SEQUENTIAL MNIST EXPERIMENT

The MNIST dataset is downloaded from the official site and then normalized by subtracting the mean value and then dividing the standard deviation of the training set. The procedure is standard and the same as many code-base. The permutation dataset is constructed by generating a random permutation of the index of each pixel(from 1 to 784) before the preprocessing procedure and then fix the random permutation during the train and test phases.

In the experiments, we use a single-layer RNN with ReLU activation. The number of the hidden node is set to 100. The size of minibatch is 64. We search the best learning rate in the range of {0.1, 0.05, 0.01, 0.05, 0.001, 0.005, 0.0001, 0.0005, 0.00001}. Each experiment runs for 400 epoch. We do not use any regularization term.

The code of this experiment is based on pytorch RNN official examples(Pytorch (2019)). The implementation of Path-SGD is adapted from the code released by the author(Neyshabur (2019)).

## A.2 LANGUAGE MODELING EXPERIMENT

The first two datasets are word level. The PTB dataset consists of 929K training words, 73K validation words, and 82K test words. It has 10k words in its vocabulary. Wikitext-2 is roughly twice the size of PTB dataset, with 2 million training words, 218k validation words, 245k test words and a vocab size of 33k. The last two datasets are character level. The PTBc dataset consists of 5017k training character, 393k validation character, and 442k test character. It has 50 alphabets in its vocabulary. The enwik8 dataset has 100M characters from Wikipedia with an alphabet size of 27. We split 90M characters for training set, 5M characters for validation set and 5M test set.

For all tasks, we train a stacked ReLU RNN with 4 layers. The embedding size is 400 and the hidden size is 1550. The learning rate is turned in the range {10, 7.5, 5, 2.5, 1, 0.75, 0.5, 0.1}. The weight decay is turned in the range { 1e-5, 1e-6, 1e-7 }. According to the (Merity et al. (2018; 2017)), we use dropouts for the input, hidden and output layer. Also, we employ weight drops. We run 750 epochs for PTB, Wikitext-2 and PTB-c experiments and run 100 epochs for enwik8 experiment which is enough for the convergence of all experiments. We identify the basis path for the RNN modules of the model and optimize the RNN modules in its PSI-space using $\mathcal{G}$-SGD algorithms. Similarly, Path-SGD is also implemented in the RNN modules. Other parameters are optimized using traditional SGD algorithms. Hyperparameters are list in the Table 3

| Hyperparameter | Value |
|---|---|
| Dropouts | 0.4 |
| Layer numbers | 4 |
| Embedding size | 400 |
| Hidden size | 1550 |
| Batch size | 40/128 |
| Gradient clip | 0.25 |
| Dropoute* | 0.1 |
| Dropouth* | 0.2 |
| Dropouti* | 0.4 |
| Wdrop* | 0.1 |

Table 3: Hyperparameters for the language modeling experiments. The batch size is 40 for the word level datasets and 128 for the character level datasets. The last four hyperparameters with * is corresponding to the implementation of the Salesforce Language Model Toolkit and the details can be find in salesforce (2019)

Performance is evaluated using the perplexity (PPL) metric for the word level datasets and bits-per-character (BPC) metric for the character level datasets. The code of this experiment is adapted from the Salesforce Language Model Toolkit(salesforce (2019)).

The implementation is based on the Salesforce Language Model Toolkitsalesforce (2019) and we only change the optimizer.

### A.3 RECURRENT CNN EXPERIMENT

The CIFAR-10, CIFAR-100 and MNIST datasets are downloaded from the official site and then normalized by subtracting the mean value and dividing the standard deviation of the training set. The code of this experiment is based on this repository (Github, 2019) which is consistent with the previous paperLiang & Hu (2015).

The RCNN network has 5 layers. There is one RCNN block in each layer and each RCNN block unfolds 3 times. As many recent code-base suggested, we using BN hear instead of LRN(Github, 2019; 2018) and removing dropout layers except the last linear layer (Github, 2019). We show all the testing error rate in the table 4.

The initial learning rate of both baseline and our method is searched in then range { 0.5, 0.2, 0.1, 0.07, 0.05, 0.01 } and then divided by 10 after 1/2, 3/4 and 7/8 epoch of all epochs. Momentum parameter is set to 0.9. Weight decay is set to 0.0001. Dropout rate is set to 0.5. Batch size is 64. Each experiment is run for 200 epoch and the test accuracy of the last epoch is reported.

Table 4: Classification error rate(%) of experiments on recurrent convolutional neural networks.

| | MNIST | | | CIFAR-10 | | | CIFAR-100 | | |
| --- | --- | --- | --- | --- | --- | --- | --- | --- | --- |
| | RCNN-32 | RCNN-64 | RCNN-96 | RCNN-96 | RCNN-128 | RCNN-196 | RCNN-96 | RCNN-128 | RCNN-196 |
| $\mathcal{G}$-SGD (our method) | **0.36** | **0.31** | **0.31** | **7.38** | **6.61** | **6.52** | **32.02** | **29.64** | **28.78** |
| SGD ( Liang(2015) ) | 0.42 | 0.32 | 0.31 | 7.37 | 7.24 | 7.09 | 34.18 | 32.59 | 31.75 |
| SGD (our implementation) | 0.38 | 0.37 | 0.39 | 7.47 | 7.53 | 7.54 | 33.34 | 31.51 | 29.67 |

## B ALGORITHMS

### B.1 DERIVATION OF THE ALGORITHM

In this section, we review the process of Inverse-Chain-Rule and Weight-Allocation designed in the work Meng et al. (2018) and apply it to RNN.

Denote the non-skeleton weight as $w_{ns}$ and the skeleton weight as $w_s$. We select the skeleton weight in the first layer as special skeleton weights and denote as $w_{s_0}$ and the non-skeleton weight in this layer is denoted as $w_{ns_0}$. Denote all the basis path set as $\mathcal{P}_b$. Denote the basis path that pass the non-skeleton weight $w_{ns}$ as $p_{ns}$ and the basis path that compose by all-skeleton weight and pass the skeleton weight $w_{s_0}$ as $p_{s_0}$. Note that there is a 1-1 mapping between non-skeleton weights and the basis paths in set $\mathcal{P}_{ns}$. Similarly, there is also a 1-1 mapping between skeleton weights $w_{s_0}$ and the basis paths in set $\mathcal{P}_{s_0}$. Here, for simplicity, we can view recurrent weight as a non-skeleton weight $w_{ns}$. Thus $\mathcal{P}_{b_2} \in \mathcal{P}_{ns}$.

Inverse-chain-rule calculates the gradient with respect to basis paths according to the following equation:

$$\left( \frac{\partial L}{\partial w_1}, \cdots, \frac{\partial L}{\partial w_m} \right) = \left( \frac{\partial L}{\partial v_{p_1}} \cdots \frac{\partial L}{\partial v_{p_{m-H}}} \right) \cdot \begin{bmatrix} \frac{\partial v_{p_1}}{\partial w_1} & \cdots & \frac{\partial v_{p_1}}{\partial w_m} \\ \vdots & \ddots & \vdots \\ \frac{\partial v_{p_{m-H}}}{\partial w_1} & \cdots & \frac{\partial v_{p_{m-H}}}{\partial w_m} \end{bmatrix} \tag{7}$$

By solving this equation, the gradients w.r.t basis paths can be derived as:

For $v_{p_{ns}}$ we have:

$$\frac{\partial L}{\partial v_{p_{ns}}} = \frac{\partial L}{\partial w_{ns}} / \frac{\partial v_{p_{ns}}}{\partial w_{ns}}.$$

For $v_{p_{s_0}}$ we have:

$$\frac{\partial L}{\partial v_{p_{s_0}}} = \frac{\partial L}{\partial w_{s_0}} - \frac{\sum_{p \in P_b/p_{s_0}} \frac{\partial L}{\partial w_{ns}} w_{ns}}{w_{s_0}}.$$

We write the result into matrix manipulation forms and we can derive line 6 in Algorithm 2 in the main paper. The detailed derivation is put at the end of this section for clarity.

Weight-Allocation projects the update on basis paths back to the update of weight. Notation $C_{s_0}$ is the production of weights in path $p_{s_0}$ except for $w_{s_0}$. Specifically, $v_{p_{s_0}} = w_{s_0} \cdot C_{s_0}$. Notation $D_{ns}$ is the production of weights in path $p_{ns}$ except for $w_{ns}$. Specifically, $v_{p_{ns}=w_{ns} \cdot D_{ns}}$. We have:

$$\frac{v_{p_{s_0}}^t - \eta \frac{\partial L}{\partial v_{p_{s_0}}^t}}{v_{p_{s_0}}^t} = \frac{v_{p_{s_0}}^{t+1}}{v_{p_{s_0}}^t} \triangleq R(v_{p_{s_0}}^t) = \frac{w_{s_0}^{t+1} \cdot C_{s_0}^{t+1}}{w_{s_0}^t \cdot C_{s_0}^t} \tag{8}$$

$$\frac{v_{p_{ns}}^t - \eta \frac{\partial L}{\partial v_{p_{ns}}^t}}{v_{p_{ns}}^t} = \frac{v_{p_{ns}}^{t+1}}{v_{p_{ns}}^t} \triangleq R(v_{p_{ns}}^t) = \frac{w_{ns}^{t+1} \cdot D_{ns}^{t+1}}{w_{ns}^t \cdot D_{ns}^t} \tag{9}$$

Solving this equation, the update rule for weight can be derived as:

$$w_{s_0}^{t+1} = w_{s_0}^t \cdot R(v_{p_{s_0}}^t) \tag{10}$$

$$w_{ns}^{t+1} = \frac{w^t - \eta_t \cdot \frac{\partial L}{\partial v_{p_{ns}}^t} / v_{p_{s_0}}^t}{R(v_{p_{s_0}}^t)} \tag{11}$$

$$w_{ns_0}^{t+1} = w_{ns_0}^t - \eta_t \cdot \frac{\partial L}{\partial w_{ns_0}^t} \tag{12}$$

We write the result into matrix manipulation forms and we can derive line 9-10 in Algorithm 2 in the main paper. The detailed derivation is put at the end of this section for clarity.

We give the derivation details in the following:

According to the chain rule, we have:

$$\frac{\partial L}{\partial w_{ns}} = \sum_{p \in \mathcal{P}_b} \frac{\partial L}{\partial v_p} \cdot \frac{\partial v_p}{\partial w_{ns}} = \frac{\partial L}{\partial v_{p_{ns}}} \cdot \frac{\partial v_{p_{ns}}}{\partial w_{ns}} \tag{13}$$

$$\frac{\partial L}{\partial w_{s_0}} = \sum_{p \in \mathcal{P}_b} \frac{\partial L}{\partial v_p} \cdot \frac{\partial v_p}{\partial w_{s_0}} = \frac{\partial L}{\partial v_{p_s}} \cdot \frac{\partial v_{p_s}}{\partial w_{s_0}} + \sum_{p \in \mathcal{P}_b/p_s} \frac{\partial L}{\partial v_p} \cdot \frac{\partial v_p}{\partial w_{s_0}} \tag{14}$$

According to the skeleton method, each non-skeleton weight $w_{ns}$ can only exists in one basis path $p_{ns}$. Besides, if $w_{ns}$ not in $p$, $\frac{\partial v_p}{\partial w_{ns}} = 0$ Thus Eq. (13) is right Thus, we can derive the gradient of the basis path as:

$$\frac{\partial L}{\partial v_{p_{ns}}} = \frac{\partial L}{\partial w_{ns}} / \frac{\partial v_{p_{ns}}}{\partial w_{ns}} \tag{15}$$

$$\frac{\partial L}{\partial v_{p_{s_0}}} = \left( \frac{\partial L}{\partial w_{s_0}} - \sum_{p \in \mathcal{P}/p_{s_0}} \frac{\partial L}{\partial v_p} \cdot \frac{\partial v_p}{\partial w_{s_0}} \right) / \frac{\partial v_{p_{s_0}}}{\partial w_{s_0}} \tag{16}$$

Let $s[w_{s_0}] = \sum_{p \in \mathcal{P}/p_{s_0}} \frac{\partial L}{\partial v_p} \cdot \frac{\partial v_p}{\partial w_{s_0}}$. We can then calculate the update ratio for the basis paths as:

$$R(v_{p_{s_0}}^t) \triangleq \frac{v_{p_{s_0}}^{t+1}}{v_{p_{s_0}}^t} = 1 - \eta_t \frac{\frac{\partial L}{\partial v_{p_{s_0}}^t}}{v_{p_{s_0}}^t} = 1 - \eta_t (\frac{\partial L}{\partial w_{s_0}^t} - s[w_{s_0}]) / (\frac{\partial v_{p_{s_0}}^t}{\partial w_{s_0}^t} \cdot v_{p_{s_0}}^t) \tag{17}$$

$$R(v_{p_{ns}}^t) \triangleq \frac{v_{p_{ns}}^{t+1}}{v_{p_{ns}}^t} = 1 - \eta_t \frac{\frac{\partial L}{\partial v_{p_{ns}}^t}}{v_{p_{ns}}^t} = 1 - \eta_t \frac{\partial L}{\partial w_{ns}^t} / (\frac{\partial v_{p_{ns}}^t}{\partial w_{ns}^t} \cdot v_{p_{ns}}^t) \tag{18}$$

$$\tag{19}$$

Above equation is right according to the gradient decent update rule of the path value $v_p^{t+1} = v_p^t - \eta_t \cdot \frac{\partial L}{\partial v_p^t}$

Finally, we project the new basis path value into the weight. Notation $C_{s_0}$ is the production of weights in path $p_{s_0}$ except for $w_{s_0}$. Specifically, $v_{p_{s_0}} = w_{s_0} \cdot C_{s_0}$. Notation $D_{ns}$ is the production of weights in path $p_{ns}$ except for $w_{ns}$. Specifically, $v_{p_{ns}} = w_{ns} \cdot D_{ns}$.

$$\frac{w_{s_0}^{t+1}}{w_{s_0}^t} \cdot \frac{C_{s_0}^{t+1}}{C_{s_0}^t} = \frac{v_{p_{s_0}}^{t+1}}{v_{p_{s_0}}^t} \tag{20}$$

$$w_s^{t+1} = R(v_{p_{s_0}}^t) \cdot w_{s_0}^t \cdot \frac{C_{s_0}^t}{C_{s_0}^{t+1}} \tag{21}$$

$$\frac{w_{ns}^{t+1}}{w_{ns}^t} \cdot \frac{D_{ns}^{t+1}}{D_{ns}^t} = \frac{v_{p_{ns}}^{t+1}}{v_{p_{ns}}^t} \tag{22}$$

$$w_{ns}^{t+1} = R(v_{p_{ns}}^t) \cdot w_{ns}^t \cdot \frac{D_{ns}^t}{D_{ns}^{t+1}} \tag{23}$$

Eq. (20) and ( 22 ) is the definition of the basis path. Note that we only update the skeleton weight in the special layer $w_{s_0}$ and the non-skeleton weight $w_{ns}$. Therefore, $\frac{C_{s_0}^t}{C_{s_0}^{t+1}} = 1$. For the non-skeleton weight in the special layer, $\frac{D_{ns}^t}{D_{ns}^{t+1}} = 1$. For the non-skeleton weight not in the special layer, $\frac{D_{ns}^t}{D_{ns}^{t+1}} = 1/R(v_{p_{s_0}}^t)$.

## B.2 NOTATIONS EXPLANATION IN THE ALGORITHM

Table 5: Notations

| notation | object |
|---|---|
| zeros_like(w) | Return a all-zero matrix whose size is the same as w |
| % | Modulus operator |
| shape(w) | Return the size of matrix w |
| w.sum(0) | Return sum of the matrix w along axis 0. If w is a weight matrix of NN , this operation sums the weight that connect to the same input and returns the vector with the same size as input node size. |

## B.3 COMPUTATIONAL COST

Here, we provide the training time per epoch of our algorithm and the baselines in Table 6. We implement all experiments on a single GPU. It shows that the real training time of G-SGD is also comparable with that of plain SGD.

Table 6: Training Time of 1 epoch (seconds) in different experiments

| | PTB | Wikitext-2 | PTB-c | enwik8 | s-MNIST-28 | s-MNIST-98 | C10-RCNN96 | C10-RCNN128 | C10-RCNN160 |
|---|---|---|---|---|---|---|---|---|---|
| Path-SGD | 53.9 | 90.2 | 79.8 | 1407 | 10.2 | 13.5 | - | - | - |
| SGD | 31.4 | 66.0 | 61.5 | 1054 | 8.5 | 10.9 | 56.0 | 74.0 | 125.5 |
| **G-SGD** | **33.9** | **68.8** | **63.8** | **1103** | **9.2** | **11.5** | **60.9** | **79.8** | **134.9** |
| Extra time cost | 2.5 | 2.8 | 2.3 | 49 | 0.7 | 0.6 | 4.9 | 5.8 | 9.4 |
| time cost increase rate | 8.0% | 4.2% | 3.7% | 4.6% | 8.2% | 5.5% | 8.7% | 7.8% | 7.5% |

## B.4 STACK RNN AND CONVOLUTIONAL LAYER

For stack RNN, there are multiple layers of hidden nodes. Specifically, there are more than one $W^h$ and $W^r$. Note that the skeleton method is also held. Thus the Mask-Matrix-Constructor algorithms can also be applied by generating mask matrix for all the weight matrix $W^h$ according to the skeleton method.

For convolutional layer, we view one feature map as a hidden node as in MLP structure. So we select one of the edges in the convolutional kernel that connect two feature maps in two layers as the skeleton weights. Thus the Mask-Matrix-Constructor algorithms can construct the mask matrix by selecting one element in one convolutional weight matrix that connects one input feature maps and one output feature maps as 1.

## C    PROOF OF THEOREMS

### C.1    PROOF OF THEOREM 1

To ease the presentation of the proof, we give the path a vector representation as follows. We index the edges in the reduction graph of neural network as $e_1, \cdots, e_i, \cdots, e_m$. We define the *counting vector* for a path $p$ either in the directed graph or in the reduction graph as $C_p = (c_{p_{e_1}}, \cdots, c_{p_{e_m}})$, where $c_{p_{e_i}}$ is equal to the times that path $p$ passes the edge $e_i$. Please note that, if path $p$ is in the directed graph, $c_{p_{e_i}}$ may be larger than 1 for some $i$ because of weight sharing. Theorem 1 is proved if we show the counting vector of any path in the directed graph is a linear combination of the counting vectors of the paths in the reduction graph.

Consider an arbitrary path in the directed graph with counting vector $C_p = (c_{p1}, \cdots, c_{pm})$. For each edge $e_i$, we choose a path in reduction graph that contains this edge, which is denoted as $p(i)$ and its counting vector is $C_{p(i)}$. We claim that the term $\sum_{e_i \in p} c_{pi} \cdot C_{p(i)} - C_p$ equals to the linear combination of counting vectors of paths in reduction graph. Without loss of generality, we suppose that $p$ contains nodes and edges $N_1^I \overset{e_1}{\to} N_2^H \overset{e_2}{\to} \cdots \overset{e_k}{\to} N_{k+1}^O$, where $e_1$ is the edge connecting an input node and a hidden node and $e_k$ is the edge connecting a hidden node and an output node, and other edges are recurrent edges. For edge $e_j, j = 3, \cdots, k-1$, we choose a path $N_{i_j}^I \overset{e_{i_j}}{\to} N_{j-1}^H \overset{e_j}{\to} N_j^H \overset{e_{z_j}}{\to} N_{z_j}^O$ that contains $e_j$, where $i_j$ denotes the index of selected edge in the input-to-hidden layer for $e_j$ and $z_j$ denote the index of selected edge in the hidden-to-output layer for $e_j$. For edge $e_{j-1}$, we choose a path $N_{i_{j-1}}^I \overset{e_{i_{j-1}}}{\to} N_{j-2}^H \overset{e_{j-1}}{\to} N_{j-1}^H \overset{e_{z_{j-1}}}{\to} N_{z_{j-1}}^O$ that contains $e_{j-1}$. Furthermore, for the edge $e_1$, we select an edge connecting with $e_1$ at the hidden-to-output layer, denoted as $e_{z_1}$. And for the edge $e_k$, we select an edge connecting with $e_k$ at the input-to-hidden layer, denoted as $e_{i_k}$. We claim that the edge $e_{i_j}$ and edge $e_{z_{j-1}}$ compose a path in reduction graph because they both connect node $N_{j-1}^H$. This claim is right for all $j = 2, \cdots, k$. Thus the linear combination of the paths that contain $e_{i_j}$ and $e_{z_{j-1}}$ equals to $\sum_{e_i \in p} c_{pi} \cdot C_{p(i)} - C_p$.

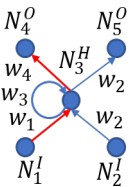

Figure 5: An example of reduction graph of RNN

Here, we use the example in Figure 5 to illustrate the above formulation. There are five weights in Figure 5, denoted as $(w_1, w_2, w_3, w_4, w_5)$. Consider the path $N_1^{I,(1)} \to N_3^{H,(1)} \to N_3^{H,(2)} \to N_3^{H,(3)} \to N_3^{H,(4)} \to N_4^{O,(4)}$ whose counting vector is $C_p = (1, 0, 3, 1, 0)$. It can be calculated as $C_p = 1 \cdot (1, 0, 0, 1, 0) + 3 \cdot (1, 0, 1, 1, 0) + 1 \cdot (1, 0, 0, 1, 0) - 4 \cdot (1, 0, 0, 1, 0)$, where $(1, 0, 0, 1, 0)$ and $(1, 0, 1, 1, 0)$ are paths in the reduction graph.

### C.2    PROOF OF THEOREM 2

Recall the definition of the activation status $a_p(w; x) = \prod_{N_i^{H,t} \in p} \mathbb{I}(N_i^{H,t}(w; x) > 0)$, We will first prove that every indicator function of the node can be calculated by the values of paths. Then using the definition of the basis paths we can know that the values of paths can be calculated using the values of basis paths. Combine the conclusion above, we can prove the theorem.

Denote the weights connect the node with index i and the node with index j as $W_{ij}^i$, consider the hidden node $N_j^{H,(t)}$, we define the value of the node as the sum of all incoming value before the activation and denote it using lower case letter $o_j^{H,(t)}$. Formally, the value of the hidden node is

$$o_j^{H,(t)} = \sum_i W_{ji}^i x_i^{(t)} + \sum_{j'} W_{jj'}^r o_{j'}^{H,(t-1)} \mathbb{I}(o_{j'}^{H,(t-1)} > 0)$$

where $x_i^{(t)}$ is the input node of time-step $t$ and $o_{j'}^{H,(t-1)}$ is the pre-activated hidden output of time-step $t-1$. The corresponding weights $W_{ji}^i$ and $W_{jj'}^r$ is in the $W^i$ and $W^r$ respectively the same as the notations in preliminaries. For simplicity, we denote the activation status of nodes as $a(o_j^{H,(t)}) \triangleq \mathbb{I}(o_j^{H,(t)} > 0)$ where $\mathbb{I}$ is the indicator function. We use notation $sign(\cdot)$ to denote the sign function($sign(a) = 1$ if $a > 0$ and $sign(a) = -1$ if $a < 0$)

We prove the theorem by using the induction method. Specifically, we will prove that the value of hidden node can be expressed by the following formulation. And thus the activation of the hidden node can be calculated by the value of path and the sign of skeleton weights in matrices $W^o$, which we call it outgoing skeleton weight and denote it as $w_s$.

$$o_j^{H,(t)} = F(x, v_p, sign(w_s)) \cdot \frac{1}{W_{kj}^o}, \tag{24}$$

where $F(x, v_p, sign(w_s))$ is a function that only related on the input $x$, values of paths $v_p$ and the signs of outgoing skeleton weights $w_s$. $W_{kj}^o$ is also an outgoing skeleton weight that connect to the hidden node $o_j^H$ and $o_k^O$.

Firstly, we prove Equation 24 is satisfied in time-step 1. For an arbitrary hidden node with index $j$

$$o_j^{H,(1)} = \sum_i W_{ji}^i x_i^{(1)} = \frac{1}{W_{kj}^o} W_{kj}^o \sum_i W_{ji}^i x_i^{(1)}$$

$$= \frac{1}{W_{kj}^o} \sum_i x_i^{(1)} \cdot v_{N_i^{I,(1)} \to N_j^{H,(1)} \to N_k^{O,(1)}}$$

$$= F(x, v_p) \cdot \frac{1}{W_{kj}^o}$$

$$= F(x, v_p, sign(w_s)) \cdot \frac{1}{W_{kj}^o}$$

Then, using induction method, the induction assumption is that $o_{t-1}$ can be represented by the function of path value divided by the weights on the skeleton edges,

$$o_{j'}^{H,(t-1)} = F(x, v_p, sign(w_s)) \cdot \frac{1}{W_{k'j'}^o}$$

Then, we can get:

$$o_j^{H,(t)} = \frac{1}{W_{kj}^o} W_{kj}^o \cdot \left( \sum_i W_{ji}^i x_i^{(t)} + \sum_{j'} W_{jj'}^r o_{j'}^{H,(t-1)} a(o_{j'}^{H,(t-1)}) \right)$$

$$= \frac{1}{W_{kj}^o} \left( \sum_i x_i^{(t)} \cdot v_{N_i^{I,(t)} \to N_j^{H,(t)} \to N_k^{O,(t)}} + \sum_{j'} W_{kj}^o W_{jj'}^r o_{j'}^{H,(t-1)} a(o_{j'}^{H,(t-1)}) \right)$$

$$= \frac{1}{W_{kj}^o} \left( \sum_i x_i^{(t)} \cdot v_{P_{N_i^{I,(t)} \to N_j^{H,(t)} \to N_k^{O,(t)}}} + \sum_{j'} \frac{W_{kj}^o}{W_{k'j'}^o} W_{jj'}^r F(x, v_p, sign(w_s)) a(o_{j'}^{H,(t-1)}) \right)$$

$$= \frac{1}{W_{kj}^o} \left( F(x, v_p, sign(w_s)) + \sum_{j'} F(x, v_p, sign(w_s)) \mathbb{I} \left( sign(W_{kj}^o) \cdot F(x, v_p, sign(w_s)) \right) \right)$$

$$= \frac{1}{W_{kj}^o} F(x, v_p, sign(w_s))$$

The second term in third equality is from the induction assumption. The fourth equality is the definition of activation function. It is obvious that for arbitrary functions $F(x, v_p, sign(w_s))$, $\frac{W_{kj}^o}{W_{k'j'}^o} W_{jj'}^r F(x, v_p, sign(w_s))$ is also a function of input, value of paths and the sign of weight on skeleton edges. The reason is that the paths in function $F$ both contain the weight $W_{k'j'}^O$. And an arbitrary value of path in RNN multiplies the weight on the recurrent edge is also the value of a path.

So we prove that if the equation 24 is satisfied in time-step $t - 1$, it will also be satisfied in time-step $t$. And the equation is proved through the induction method.

Next, recall the definition of the activation status of paths: $a_p(w; x) = \prod_{N_i^{H,t} \in p} \mathbb{I}(o_i^{H,t}(w; x) > 0)$. By using Equation (24), we can know that the sign of $o_j^{H,(t)}$ can be calculated using the value of basis paths given the sign of weights on skeleton edges. Combine it with that the value of path can be calculated by the value of basis path, we can easily prove the theorem.

### C.3 PROOF OF THEOREM 3

As we mentioned, the structure of ReLU RNN can be decomposed into a feedforward part and the recurrent edges part. For the feedforward part, the number of basis paths has been proved to be #(weights)−#(hidden nodes) in Meng et al. (2018). According to our Skeleton Method for RNN, each recurrent weight can only be contained in one basis paths. Thus, the total number of recurrent neural networks is #(weights of MLP)+#(recurrent weights)−#(hidden nodes). Thus, we finish the proof.

## D TESTING ACCURACY CURVES FOR VISION TASKS

### D.1 SEQUENTIAL MNIST

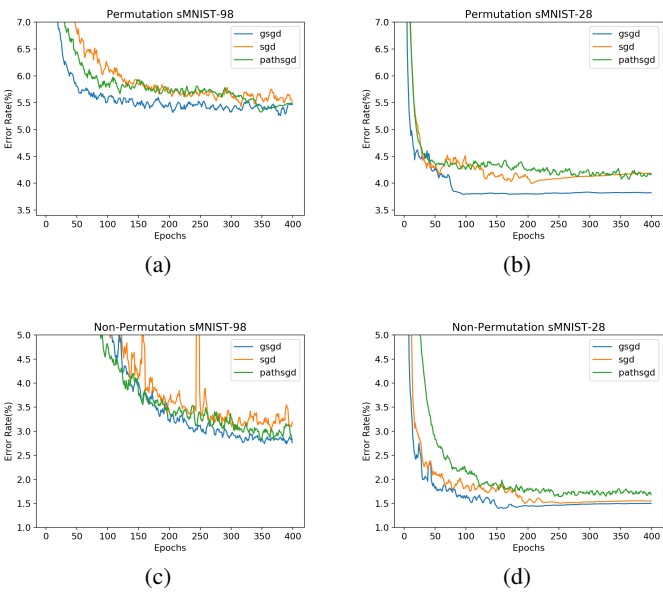

Figure 6: Test Error Rate for Sequential MNIST experiments. x axis is the epoch index, y axis is the Error Rate(%)

### D.2 RECURRENT CNN

## E ADDITIONAL $\mathcal{G}$-ADAM V.S. ADAM EXPERIMENT RESULTS ON SEQUENTIAL MNIST EXPERIMENTS

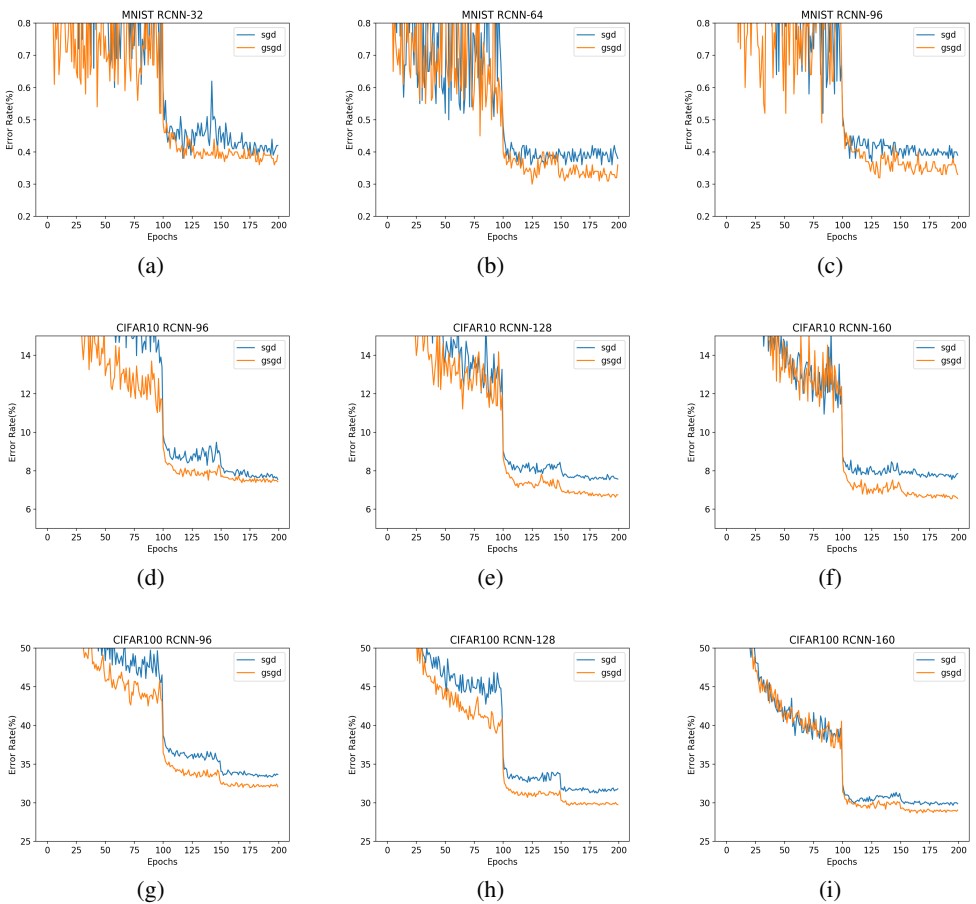

Figure 7: Test Error Rate for Recurrent CNN experiments on 3 datasets. x axis is the epoch index, y axis is the Error Rate(%)

Table 7: Averaged testing classification error rate(%) of the sequential MNIST experiment.

|  |  | $\mathcal{G}$-ADAM | ADAM |
|---|---|---|---|
| Non-Permutation | sMNIST-28 | **1.50 ± 0.07** | 1.72 ± 0.07 |
|  | sMNIST-98 | **2.49 ± 0.08** | 3.10 ± 0.10 |
| Permutation | sMNIST-28 | **3.73± 0.05** | 4.06± 0.1 |
|  | sMNIST-98 | **5.42± 0.1** | 5.57± 0.09 |

