# OpenReview forum: "Path Space for Recurrent Neural Networks with ReLU Activations"
_ICLR.cc/2020/Conference — Reject_

### Official Review · AnonReviewer3 · 2019-10-23
**Official Blind Review #3**

**Rating:** 3

**Review:**

This paper aims to improve the training of recurrent neural networks (RNN) with ReLU activations by optimizing in the path space instead of the weight space. Studies on multi-layered perceptrons (MLP) and convolutional neural networks (CNN) have shown that these architectures are positively scale invariant (PSI), however traditional SGD optimizes in the weight space that does not have this property. It has been shown in prior work that this mismatch can slow down the optimization process for SGD and it has been demonstrated that optimizing in the so called path space, which has the PSI property, can be faster and more efficient.  The authors of this paper aim to extend an existing path-space framework (G-SGD) that is used for ReLU networks to facilitate RNNs.  To tackle the challenge posed by the time-dependency of RNNs, they use a static representation, called reduction graph, to define the path space. First, they prove that any path in the original RNN graph can be easily obtained from paths in its reduction graph by simple operations. Then, they define the basis for the path space in the reduction graph, and show that basis paths are sufficient to represent the output of the RNN. They propose a method (Skeleton method) to generate a basis path in the reduction graph and specify an equivalent of the G-SGD algorithm for RNNs.  Through numerical simulations they demonstrate that (1) G-SGD for RNNs has a fairly low additional computational cost compared to SGD and (2) G-SGD for RNNs achieves better test accuracy compared with SGD and an additional path space based approach.

Even though the paper has some original contribution, due to issues with the delivery, clarity and execution of the paper I would lean to reject it at this point. I would be willing to change my decision if the authors made significant changes to the paper in aspects detailed below. The paper also has many grammatical and typographical mistakes that hinders the reader in understanding key ideas.

First,  Section 2 could be much better explained. It seems like the notation introduced for a node and its value are used interchangeably, the edges are not clearly defined and Definition 1 is extremely confusing (what is s’ an arbitrary time step or all time steps between t and t+s, where is the list of weights mentioned in the first sentence and so on).  The definition for a key concept in the paper, the value of the path, is also not clear due to the lack of sufficient definition of the graph itself.

Second, in Section 3 the definition of the reduction graph and how it is obtained from the directed graph is not clear, which is problematic since it is one of the key ideas in the paper. Clear explanation of a recurrent edge is also missing.

Third, in Section 4 terms such as ‘without recurrent edges parts’  that are used in defining the Skeleton method are very lax and needs to be defined more rigorously. It is hard to understand what the boxed steps are saying. Moreover,  in Algorithm 2 division by a matrix occurs at several points. Does this mean entry-wise division? What does the function BP(.) do? I couldn’t find it defined anywhere.

Lastly, in Section 5 it would be fair to make a comparison with adaptive optimization methods such as Adam or AdaGrad as it is done in [Neyshabur et al.: Path SGD].

I would like to provide some additional feedback that do not impact my decision, but could potentially improve the paper. In Section 2: I cannot find an explanation what an ‘unfold directed graph’ means. In eq. (3) x is used without any prior explanation/definition. In Section 3, the message around eq. (4) is not clear, how we pick p2…p5? Moreover, in the definition of basis paths P is used without explaining what it denotes.  Comments on Appendix: in Table 4 it is confusing to highlight the results corresponding to the papers method, because it suggests that these are the best results across the table, however there is a lower value in the second row. In B.1. taking derivative w.r.t p doesn’t seem to make sense, it should probably be v_p (this holds for the whole derivation). In Table 6, it would be a clearer to present relative (%) results than absolute time cost.  Comment on the proofs: they are extremely verbose and could potentially be explained with more math and less words. It is difficult to follow what is going on. There is also some notation not introduced before in eq. (28). In general, I would recommend only numbering equations that are referred to in the text.

**Experience Assessment:**

I have read many papers in this area.

**Review Assessment: Checking Correctness Of Derivations And Theory:**

I assessed the sensibility of the derivations and theory.

**Review Assessment: Checking Correctness Of Experiments:**

I assessed the sensibility of the experiments.

**Review Assessment: Thoroughness In Paper Reading:**

I read the paper at least twice and used my best judgement in assessing the paper.

---

> ### Author Response · Authors · 2019-11-15
> **Response To Reviewer #3**
>
> 1.      Thanks for your valuable suggestions on this paper. We have made the following revisions. In Section 2, we have revised the description of the graph for RNN by adding the description of edges. We also refine Definition 1 and change to use different notations to denote node and its value.
>
> 2.      Please note that the definition of the value of path was shown in the previous version and we emphasize it using boldface in the new version.
>
> 3.      In Section 3, we add a formal definition of the reduction graph in Definition 3.
>
> 4.      In Section 4, we change the “without-recurrent part” into the “forward part” and describe it before the box and revise the Skeleton method in box.
>
> 5.      For the questions about the division, the division shown in Algorithm 2 means the element-wise division. The function BP(.) returns the gradient of weights, which is calculated by using the Back-Propagation method.
>
> 6.      For your additional feedback, we revised our paper in the following aspects:
> a.      We remove the term “unfold directed graph” and use the term “directed graph” instead. We define the directed graph by defining its nodes and edges in Definition 2.1 and illustrate it in Figure1.
> b.      x in Eq.(3) means the input value. We add the its explanation before we use it.
> c.      We use Eq.(4) to illustrate that “we cannot directly optimize the values of all the paths by regarding each of them to be an independent parameter, because they are correlated with each other” as shown in the paragraph above the Eq. (4).  So, p2, …, p5 are just an example to show the dependent between paths and they are a subset of basis path. In the following part, we show that we can identify all the basis paths efficiently (skeleton method)
> d.      Table 4: Since comparing the numbers between the different implementations is unfair, we reproduce the SGD baseline and show the number in the third row.  The second row in Table 4 is borrowed from the previous paper and just for reference. In fact, some of our reproduced numbers are better than the previous work and others are worse. The meaningful comparison is between the first row and the third row and it shows our results are always better than the baseline method
> e.      Thank you for pointing out the typos  and we change p into v_p in proofs in Appendix. .
> f.       We add one row to show the relative time cost rather than the absolute value.
>
>
> Besides the above items, we have also revised other parts of the paper including proofs, numbers in tables in the appendix according to your suggestions. Please check the new version and hope you can reconsider your rate.

---

### Official Review · AnonReviewer1 · 2019-10-24
**Official Blind Review #1**

**Rating:** 3

**Review:**

Motivation: The authors motivate their work with the observation that neural networks with ReLU activations are positively scale invariant. Recent works have proposed parameter space called path space to leverage this insight for feedforward as well as CNNs, but this is not really the case for Recurrent Neural Networks (due to the recurrent structure and the parameter sharing).

Contribution:  The authors  construct path space for RNN to employ optimization algorithms in path space. The intuition is to leverage the reduction graph of RNN to removes the influence of time-steps. Reduction graph only contains information about the weight connectivity patterns and not about the time steps. Hence,  the number of paths in reduction graph is fixed (i.e independent of number of time steps.

Clarity: The clarity of the paper can be improved a bit. It might be useful to give a "top-down" introduction somewhat at the end of introduction.

Experimental Results: In order to validate the proposed method, the authors conduct experiment on sequential MNIST tasks, as well as language modelling task. The results are a bit weak in general, and improvements are very minor over the vanilla RNN baseline. It might be interesting to compare to other baselines as in PathSGD paper.





**Experience Assessment:**

I have published in this field for several years.

**Review Assessment: Checking Correctness Of Derivations And Theory:**

I assessed the sensibility of the derivations and theory.

**Review Assessment: Checking Correctness Of Experiments:**

I assessed the sensibility of the experiments.

**Review Assessment: Thoroughness In Paper Reading:**

I read the paper thoroughly.

---

> ### Author Response · Authors · 2019-11-15
> **Response To Reviewer #1**
>
> Thanks for your valuable comments about the clarity.
>
> Please note that the paragraph 5-7 in the introduction in the previous version of our paper summarize our main works and contributions. In the new version of our paper, we add a “top-down” introduction to the whole work to the end of the introduction. For your concern about the experiments, we can conclude that as the task becomes more difficult (from MNIST to CIFAR, from shallower NN to deeper NN), the improvement of our proposed optimization methods becomes larger.
> For the recurrent CNN experiments, our improvements are significant compared with the number shown in Recurrent CNN paper (Liang, Ming, and Xiaolin Hu. CVPR. 2015.).
>
> Besides, we add Adam optimizer as a new baseline. The results are shown in Table 7 in the Appendix. Additionally, we repeat each sequential MNIST experiment 3 times and list the mean and variance of each result to show our improvements are stable.

---

### Official Review · AnonReviewer2 · 2019-11-04
**Official Blind Review #2**

**Rating:** 1

**Review:**

This paper proposes a parameter space, called path space for RNNs with ReLU activation. For the construction of the path space, this paper utilises a reduction graph approach to minimise the difficulty brought by the parameter-sharing scheme in RNNs. Furthermore, the authors propose a Skeleton method for the efficient identification of basis path for RNNs in reduction graph.
The G-SGD approach used in this work is not new and has been initially applied by Meng et al., 2018.
Meng et al., 2018 have primarily introduced G-space for neural networks and designed SGD in G-space (G-SGD) to optimise the value vector of the basis paths of neural networks.  The main difference between this manuscript and previous works (Meng et al., 2018 and Neyshabur et al., 2016.) is that the authors apply G-SGD to RNNs instead of MLPs/CNNs.
The reviewer is not sure if this marginal difference is sufficient to get the paper accepted in ICLR. The structure, the methodology, and the content of this paper is highly similar to the work published by Meng et al., 2018.
This paper claims to obtain significantly more effective RNN models than using optimization methods in the weight space without providing any statistically significant measures.
The authors could have added a visual performance comparison (e.g., training loss/epoch curve and test error/epoch curve) between the RNN G-SGD and other state-of-the-art approaches.
The theorems and their proofs are fine (similar to Meng et al. (2018)).


**Experience Assessment:**

I do not know much about this area.

**Review Assessment: Checking Correctness Of Derivations And Theory:**

I assessed the sensibility of the derivations and theory.

**Review Assessment: Checking Correctness Of Experiments:**

I assessed the sensibility of the experiments.

**Review Assessment: Thoroughness In Paper Reading:**

I read the paper at least twice and used my best judgement in assessing the paper.

---

> ### Author Response · Authors · 2019-11-15
> **Response to Reviewer #2**
>
> First, it is unfair to give this paper 1 point by saying that this paper is highly similar to the previous paper. The differences between our paper and the previous work G-SGD are stated in paragraph 4 in the introduction and the contribution of our work is shown in paragraph 5-7 in the introduction.  Specifically, due to the recurrent structure and the parameter sharing, this work is significantly different from the previous work that considering the MLP/CNN (Meng et al., 2019). In RNN, all paths are related not only to the RNN structure but also to the sequence length of the data. This property makes the basis path definition and identification problematic since we want to define and identify the maximal independent group from the undetermined number of paths. To solve this problem, we propose using reduction graph as a tool and we prove that the basis path of reduction graph is equivalent to the basis path of RNN. Since all the paths in the reduction graph are not related to the sequence length of the input data, we can achieve our goal and construct the path space for RNN.
>
> Second, for your concerns about experiments, we present the experiment results with different random seeds to verify our improvement is significant.[QM1]  For the experiments for RCNN, our improvements are significant compared with the number shown in the original Recurrent CNN paper (Liang, Ming, and Xiaolin Hu. CVPR. 2015.). We can conclude that as the task becomes more difficult (from MNIST to CIFAR, from shallower NN to deeper NN), the improvement of our proposed optimization methods becomes larger.
>
> Thanks for your suggestions, we have added curves for vision task in Section D in the appendix.
>
> We hope we have addressed your concerns and you can reconsider the rate.

---

### Author Response · Authors · 2019-11-15
**Response to reviewers**

We sincerely thank all the reviewers for their efforts on our paper and their helpful suggestions!

We have uploaded a new version of our paper, and the main revisions are listed below:

1.	We add a “top-down” introduction of our work to the end of the introduction.

2.	We add some experiments. Specifically, we repeat each sequential MNIST experiment 3 times and show the mean and variance of each experiment in Table 1 and 7. Additionally, we add the Adam  as our baseline and compare our G-Adam (Adam in path space) optimizer  with Adam. The results are shown in Table 7 in the Appendix.

3.	According to suggestions from Reviewer #3,
(1) we revise the definition of path in RNN and the notation for clarity;
(2)  we add a formal definition of reduction graph in Definition 3;
(3)we add more details about the Skeleton Method.

4.	We fix some typos and grammar errors and proofread the proofs in Appendix.

---

### Decision · Program_Chairs · 2019-12-19

**Decision:**

Reject

**Comment:**

The scores of the reviewers are just far to low to warrant an acceptance recommendation from the AC.